# Therapeutic Advances and Challenges in the Management of HER2-Positive Gastroesophageal Cancers

**DOI:** 10.3390/diseases10020023

**Published:** 2022-04-19

**Authors:** Jeremy Chuang, Samuel Klempner, Kevin Waters, Katelyn Atkins, Joseph Chao, May Cho, Andrew Hendifar, Alexandra Gangi, Miguel Burch, Pareen Mehta, Jun Gong

**Affiliations:** 1Department of Medical Oncology and Therapeutics Research, City of Hope Comprehensive Cancer Center, 1500 E Duarte Rd, Duarte, CA 91010, USA; jechuang@coh.org (J.C.); jchao@coh.org (J.C.); 2Mass General Cancer Center and Department of Medicine, Massachusetts General Hospital, 55 Fruit St, Boston, MA 02114, USA; sklempner@partners.org; 3Department of Pathology and Laboratory Medicine, Cedars-Sinai Medical Center, 8700 Beverly Boulevard, Los Angeles, CA 90048, USA; kevin.waters@cshs.org; 4Department of Radiation Oncology, Cedars-Sinai Medical Center, 8700 Beverly Boulevard, Los Angeles, CA 90048, USA; katelyn.atkins@cshs.org; 5Division of Hematology and Oncology, UCI Health Chao Family Comprehensive Cancer Center, 101 The City Drive South, Building 23, Orange, CA 92868, USA; mayc5@hs.uci.edu; 6Department of Hematology/Oncology, Cedars-Sinai Medical Center, 8700 Beverly Boulevard, Los Angeles, CA 90048, USA; andrew.hendifar@cshs.org; 7Division of Surgical Oncology, Cedars-Sinai Medical Center, 8700 Beverly Boulevard, Los Angeles, CA 90048, USA; alexandra.gangi@cshs.org; 8Department of Surgery, Cedars-Sinai Medical Center, 8700 Beverly Boulevard, Los Angeles, CA 90048, USA; miguel.burch@cshs.org; 9Department of Radiology, Cedars Sinai Medical Center, 8700 Beverly Boulevard, Los Angeles, CA 90048, USA; pmehta@theangelesclinic.org; 10Samuel Oschin Comprehensive Cancer Institute, Cedars-Sinai Medical Center, 8700 Beverly Boulevard, Los Angeles, CA 90048, USA

**Keywords:** gastroesophageal cancer, trastuzumab, HER2, next-generation sequencing, circulating tumor DNA

## Abstract

Gastroesophageal cancer is one of the most common cancers in the world, with a high rate of mortality. While there has been significant progress over the past decade, particularly with the addition of anti-HER2 therapies to platinum-based chemotherapy agents in the advanced setting, the prognosis remains poor and the treatment options for this disease entity remain limited. In this review, we discuss the current therapeutic landscape for HER2-positive gastroesphageal cancer and the seminal clinical trials that have shaped our approach to this disease entity. In addition, we highlight some of the challenges to the understanding and management of this disease, specifically discussing the breadth of molecular diversity and intratumoral heterogeneity of HER2 expression that impact the clinical efficacy and prognosis. Furthermore, we discuss the potential role of next-generation sequencing (NGS) and circulating-tumor DNA (ctDNA) as complementary tools to immunohistochemistry (IHC) and fluorescent in-situ hybridization (FISH) to guiding clinical decision making. Finally, we highlight promising clinical trials of new treatment regimens that will likely reshape the therapeutic approach to this disease entity.

## 1. Introduction

Gastroesophageal cancers remain one of the most common cancers in the United States, with an estimated incidence of approximately 45,820 cases in 2021 [1]. Despite the decreasing incidence and mortality from the disease over the past decade, the cost and burden on the healthcare system has increased substantially [2]. The improved outcomes and rise in costs may be in part related to the utilization of targeted agents and checkpoint inhibitors in the management of advanced gastroesophageal adenocarcinoma (GEA). While the therapeutic landscape has changed significantly over the past decade, the prognosis for advanced gastroesophageal malignancy remains poor and highlights the importance of research and development for novel therapies and strategies [3]. Currently, systemic chemotherapy remains the cornerstone of management for advanced GEA with multidrug combinations including fluoropyrimidine and platinum-based regimens with or without HER2-targeted agents and checkpoint inhibitors.

HER2, also known as *erythroblastosis oncogene B2* or *ERBB2*, is an epidermal growth factor receptor (EGFR) tyrosine kinase receptor on the cell membrane that plays a critical role in cell proliferation. Overexpression of HER2, defined as 3+ by immunohistochemistry (IHC) staining or positivity by fluorescence in situ hybridization (FISH; HER2/chromosome enumeration probe 17 [CEP17] ratio ≥ 2.0 or average HER2 copy number ≥ 6.0 signals per cell), is an important predictive biomarker with positivity rates that vary depending on anatomic location and histology [4,5]. Gastroesophageal junction tumors have a higher HER2-positivity rate (32%) compared to tumors in the stomach (21%). Furthermore, tumors with intestinal subtype have a higher propensity for HER2 overexpression (32%) compared to tumors with diffuse subtype histology (6%). Clinically, HER2 overexpression guides the use of the monoclonal antibody trastuzumab as an adjunct to systemic chemotherapy in the first-line setting and the antibody conjugate fam-tastuzumab derutxecan-nxki in subsequent lines of therapy. In this review, we will detail the landmark trials that support the use of HER2-targeted agents in GEA and highlight promising novel therapies for the future.

## 2. HER2-Targeted Agents in the First-Line Setting

HER2-targeted agents came into prominence for GEA in 2010, with the phase III ToGA clinical trial demonstrating an improvement in overall survival with the addition of trastuzumab to systemic chemotherapy (Table 1) in patients with locally advanced, recurrent, or metastatic gastric or GEJ adenocarcinoma demonstrating HER2 overexpression [6]. The study enrolled 594 patients to either trastuzumab with chemotherapy (capecitabine/cisplatin or 5-fluorouracil/cisplatin given every 3 weeks for 6 cycles) or chemotherapy alone. The majority of the patients harbored tumors in the stomach (80%). Overall HER2 positivity rate was 22.1%, with higher rates seen in tumors with Lauren intestinal subtype compared to diffuse subtype tumors (31.8% vs. 6.1%). Furthermore, gastroesophageal tumors had higher HER2-positivity rates compared to tumors in the distal or gastric body (32.2% vs. 21.4%). The clinical trial met its primary endpoint demonstrating a significant improvement to overall survival in the experimental arm with an overall survival (OS) of 13.8 vs. 11.1 months (Hazard ratio (HR), 0.74; *p* = 0.0046). Subgroup analysis demonstrated that tumors with HER2 IHC 2+ and FISH positive or IHC 3+ appeared to derive the greatest benefit with a median OS of 16.0 vs. 11.8 months (HR, 0.65; 95% confidence interval (CI) 0.51–0.83, *p* = 0.036).

The 2017 HELOISE phase III clinical trial sought to build upon the findings in the ToGA study by exploring higher dosing of trastuzumab (8 mg/kg loading dose, followed by 6 mg/kg vs. 10 mg/kg every 3 weeks) in comparison to standard dosing (trastuzumab 10 mg/kg every 3 weeks) in combination with systemic chemotherapy (cisplatin 80 mg/m^2^ plus capecitabine 800 mg/m^2^ twice per day in cycles 1 to 6) [7]. However, in this study, the authors did not find a statically significant difference in median OS between the two arms (12.5 vs. 10.6 months *p* = 0.2401). Furthermore, there were no significant differences in the safety profiles of the two different doses. Therefore, the current and optimal dosing for trastuzumab remains the same as the one studied in the ToGa trial.

The phase III LOGiC clinical trial explored whether the small molecular tyrosine kinase inhibitor (TKI) lapatinib would provide clinical benefit in patients with locally advanced or metastatic esophageal, gastric, or GEJ adenocarcinoma [8]. 545 patients with HER2-positive locally advanced or metastatic gastroesophageal adenocarcinoma were randomized to receive either lapatinib in combination with capecitabine/oxaliplatin or capecitabine/oxaliplatin alone. The primary endpoint was not met with no difference in median OS (12.2 vs. 10.5 months, HR = 0.91; 95% CI 0.73–1.12, *p* = 0.3492). However, a preplanned subgroup analysis demonstrated an improved OS in younger patients and those of Asian ethnicity. There was also no improvement in progression-free survival (PFS) (6.0 vs. 5.4 months, *p* = 0.0381). The experimental arm did demonstrate higher rates of diarrhea (58% vs. 29% for all grades) and grade ≥ 3 diarrhea (12% vs. 3%). These findings do not support the use of lapatinib in the front-line setting for advanced GEA.

Studies have explored whether the addition of pertuzumab, a monoclonal humanized immunoglobin that targets HER2, may provide clinical benefit to patients with HER2-positive GEAs, as is seen in breast cancer. A phase IIa study randomized 30 patients to receive either pertuzumab 840 mg for cycle 1 and 420 mg every 3 weeks for cycles 2–6 (Arm A) or pertuzumab 840 mg every 3 weeks for six cycles (Arm B) to explore the pharmacokinetic and safety profile [9]. During the study, the patients also received trastuzumab, cisplatin, and capecitabine for 6 cycles, followed by trastuzumab every 3 weeks. Given the safety and pharmacokinetic data, pertuzumab 840 mg every 3 weeks was selected to be evaluated in the phase III JACOB clinical trial [10]. In this study, 780 patients with HER2-positive metastatic GEA were randomly assigned to receive either chemotherapy (cisplatin 80 mg/m^2^ every 3 weeks intravenously, oral capecitabine 1000 mg/m^2^ twice a day [2000 mg/m^2^ every 24 h] for 28 doses every 3 weeks, or 5-fluorouracil 800 mg/m^2^ every 24 h intravenously [120 h continuous infusion] every 3 weeks) with trastuzumab and pertuzumab or chemotherapy with trastuzumab alone. After a median follow up of 24.4 months, the primary endpoint of overall survival was not met (17.5 vs. 14.2 months, HR = 0.84, *p* = 0.057). However, there was a significant increase in median PFS (8.5 vs. 7.0 months, HR = 0.73, 95% CI 0.62–0.86, *p* = 0.026). Given the findings in the JACOB trial, pertuzumab is not part of the standard of care for locally advanced or metastatic HER2-positive GEAs.

Recent studies explored the addition of checkpoint inhibitors to HER2 targeted agents in addition to systemic chemotherapy. A phase Ib/II PANTHERA trial from Korea demonstrated promising efficacy of frontline pembrolizumab, trastuzumab, and chemotherapy (capecitabine 1000 mg/m^2^ bid D1-14 and cisplatin 80 mg/m^2^ D1 every 3 weeks) in HER2-positive advanced gastric and gastroesophageal cancer irrespective of PD-L1 status [11]. A total of 43 patients enrolled were followed for a median of 18.2 months and found to have an ORR of 76.7% (complete response (CR) 16.3%, partial response (PR) 60.5%). Median OS was 19.3 months (95% CI 16.5-not reached (NR)) and median PFS was 8.6 months (95% CI 7.2–16.5). PD-L1 status (55.3% of patients ≥ CPS 1 and 13.2% of pts ≥ CPS 10) did not correlate with survival. Of interest in the study was the use of a trastuzumab biosimilar for the experimental arm. Given the rising costs in oncology, this development will have significant implications on increasing accessibility and reducing the cost of care for cancer patients.

Other promising studies have also evaluated the role of immunotherapy with HER2 targeted agents. A phase II study enrolled 37 patients with HER-positive metastatic GEA to receive systemic chemotherapy with trastuzumab and pembrolizumab after an induction cycle of trastuzumab and pembrolizumab only. After a median follow up of 13 months, the primary endpoint of median PFS was achieved with 26 of 37 (70%; 95% CI 54–83%) patients being progression-free at 6 months. The most common adverse event was neuropathy, which 36/37 (97%) patients experienced. Common grade ≥ 3 adverse events included lymphocytopenia, anemia, and decreased electrolytes. These findings demonstrated promising clinical efficacy and a manageable safety profile for additional study in a phase III clinical trial. Interim analysis of the phase III KEYNOTE-811 study led to accelerated approval of pembrolizumab with first-line platinum-based chemotherapy and trastuzumab for HER2-positive gastric cancer on 5 May 2021, irrespective of PD-L1 status [12]. Amongst the first 264 patients with locally advanced or metastatic HER-positive GEA enrolled in this study, 133 patients were randomized to the standard of care (5-flurouracil and cisplatin or capecitabine and oxaliplatin) with or without pembrolizumab given 200 mg every 3 weeks. Primary endpoints included OS and PFS. After a median follow up of 12 months, the overall response rate (ORR) was 74.4 vs. 51.9% (difference, 22.7 percentage points [95% CI, 11.2–33.7], *p* = 0.00006). A total of 11.3% of patients were able to achieve a complete response compared to 3.1% in the control arm. Toxicity profile was comparable in each arm with a grade ≥ 3 rate of 57.1% in the experimental arm vs. 57.4% in the control arm. These findings support the addition of pembrolizumab to the standard of care for HER2-positive advanced GEA.

## 3. HER2-Targeted Agents in the Second Line and Beyond

The development of HER2-targeted agents has made its way into treatment-refractory settings in advanced GEA (Table 2). Given the promising role of trastuzumab emtansine (T-DM1), an antibody-drug conjugate with a microtubule inhibitor that induces an antibody-dependent cellular cytotoxicity, in breast cancer, prospective studies have explored its potential role in GEAs [13,14]. GATSBY was a phase II/III clinical study that enrolled patients with previously treated HER2-positive locally advanced or metastatic GEA [15]. In the first part of the study, 182 patients were assigned to receive either intravenous T-DM1 (3.6 mg/kg every 3 weeks or 2.4 mg/kg weekly) or physician’s choice of a taxane. An interim analysis determined T-DM1 2.4 mg/kg weekly as the dose for the second stage of the study, wherein an additional 233 patients were assigned 2:1 to receive either T-DM1 or physician’s choice of a taxane. T-DM1 2.4 mg/kg weekly was demonstrated to be safe and tolerable [16]. The primary endpoint was not met in this study, with a median overall survival of 7.9 vs. 8.6 months (HR, 1.15; 95% CI 0.87–1.51, one-sided *p* = 0.86). The conclusion of this study was that T-DM1 was not superior to taxanes in the second line and beyond. It is important to note that repeat HER2 testing was not required after progression. Therefore, it is plausible that there was a lack of clinical efficacy observed due to the given tumors without HER2-positivity being included in the analysis of the study.

Lapatinib has also been explored as a potential therapeutic option for the second-line setting in the phase III TyTAN study [17]. 273 patients with previously treated HER2-positive metastatic GEA were randomized to received either lapatinib 1500 mg daily plus weekly paclitaxel 80 mg/m^2^ or paclitaxel alone. Because the trial enrollment occurred during the time period in which ToGA was being adopted, relatively few patients had received a trastuzumab-based regimen in the first-line setting. Nonetheless, the primary endpoint was not reached and median OS was 11.0 vs. 8.9 months (HR, 0.84; 95% CI 0.64–1.11; *p* = 0.1044). In subgroup analyses, it did appear that the experimental arm demonstrated improved clinical efficacy in tumors with IHC 3+ compared to IHC 0/1+/2+ patients. These findings suggested a possible benefit for tumors with enriched HER2 overexpression.

The T-ACT phase II clinical trial explored whether there would be a clinical benefit of continuing trastuzumab beyond progression in patients with refractory HER2-positive locally advanced or metastatic GEA treated with fluoropyrimidine, platinum, and trastuzumab in the first-line setting [18]. In this study, 91 patients were randomized to receive paclitaxel (80 mg/m^2^ on days 1, 8, and 15 every 4 weeks) or paclitaxel plus trastuzumab (8 mg/kg loading dose with 6 mg/kg every 3 weeks). There was no difference in the primary endpoint of median PFS (3.19 vs. 3.68 months, *p* = 0.334) or secondary endpoint of median OS (9.95 vs. 10.20 months, *p* = 0.199). Of interest, the investigators collected new tumor-biopsy samples in a select number of patients (*n* = 16) and found that 11/16 (69%) patients no longer demonstrated HER2 positivity. These findings highlight the importance of a repeat tumor biopsy upon progression to determine HER2 status to guide appropriate therapy upon progression of disease.

Several studies assessed the clinical efficacy of trastuzumab deruxtecan in previously treated HER2-positive gastric cancer. The phase II DESTINY-Gastric01 clinical trial that enrolled 187 patients predominantly from South Korea and Japan who had received two prior lines of therapy including trastuzumab to receive either trastuzumab deruxtecan or physician’s choice of chemotherapy [19]. The primary endpoint of ORR was 51% in the experimental arm vs. 14% in the control group (*p* < 0.001). Median OS was significantly improved 12.5 vs. 8.4 months (HR 0.59; 95% CI 0.39–0.88; *p* = 0.01). The most common grade ≥ 3 adverse events with trastuzumab deruxtecan were cytopenias, including neutropenia, anemia, and leukopenia. A total of 12 patients in the experimental arm also experienced treatment-related interstitial lung disease and pneumonitis, with 3/12 patients experiencing grade ≥ 3 events. These findings led to the accelerated FDA approval on January 15 2021 for the treatment of HER2-positive advanced gastroesophageal malignancy after progression on a trastuzumab-based chemotherapy.

Preliminary data of the single-arm phase II Destiny-Gastric 02 presented on 17 September 2021 at ESMO found trastuzumab deruxtecan to demonstrate clinical benefit and a manageable toxicity profile as a second-line therapy in HER2+ unresectable and metastatic gastroesophageal cancer in a predominantly Western population [20]. In this study, 79 patients received trastuzumab deruxtecan 6.4 mg/kg intravenously every 3 weeks, and the study results demonstrated an overall response rate in 30 patients (38%, 95% CI, 27.3–49.6). A total of 3 patients (3.8%) were able to achieve a complete response, while partial response was seen in in 27 patients (34.2%), stable disease in 34 patients (43.0%), and progressive disease was seen in 13 patients (16.5%). Median PFS was 5.5 months (95% CI, 4.2–7.3). Grade ≥ 3 adverse events were observed in 21 patients (26.6%), with the most common treatment-related adverse events including nausea (58.2%), fatigue (36.7%), and vomiting (32.9%). While these findings are preliminary, the initial findings are promising and demonstrate the potential of trastuzumab deruxtecan as a therapeutic option in the second-line setting with data to support its use in the West.

## 4. Discussion

The management of HER2-positive gastroesophageal cancer continues to evolve with the development of novel therapies and strategies; however, there remain significant questions and challenges to the management of this disease entity. While the continuation of trastuzumab upon progression is an effective strategy in breast cancer, that approach remains controversial in gastric cancer [21]. A retrospective study of 104 patients explored the role of continuing trastuzumab after progression with a platinum-based chemotherapy and trastuzumab in HER2-positive advanced GEAs [22]. The continuation of trastuzumab beyond progression resulted in significantly improved median OS (12.6 vs. 6.1 months; *p* = 0.001) and PFS (4.4 vs. 2.3 months; *p* = 0.002) compared to patients who discontinued therapy. However, recent prospective studies have been mixed regarding the role of continuing anti-HER2 agents beyond progression. As previously discussed, the T-ACT phase II clinical trial did not demonstrate a clinical benefit in continuing trastuzumab beyond progression in patients previously treated with a trastuzumab-based regimen in the first-line setting [18]. It is worth emphasizing that of the small sample of patients who did receive a repeat biopsy upon progression, there appeared to be a significant loss of HER2-positivity status in 11/16 patients (69%), which may potentially explain the lack of clinical efficacy of continuing trastuzumab in this study. Nonetheless, these findings emphasize the importance of clarifying HER2 status to guide selection of appropriate therapy at time of progression.

While the continuation of trastuzumab upon progression has not yet proven to provide clinical benefit in HER2-positive gastroesophageal cancer, there still may be a role for anti-HER2 agents in subsequent lines of therapy. In the prospective DESINTY-gastric 01/02 clinical trials, trastuzumab deruxtecan appeared to provide robust response rates and also an improvement in median overall survival [19,20]. It has been postulated that trastuzumab deruxtecan is able to provide a robust clinical response and benefit via the “bystander killing effect”, whereby trastuzumab deruxtecan is internalized by HER2-positive cancer cells and the cytotoxic payload is then transferred to adjacent tumor cells with subsequent apoptosis of cancer cells [23]. Therefore, the antitumor activity of trastuzumab deruxtecan may be driven more by the cytotoxic payload of chemotherapy rather than the targeting of HER2, which acts more like an entry point into cancer cells. Ultimately this may allow for this agent to overcome the significant intratumoral heterogeneity of HER2 expression, which has been recognized within gastroesophageal cancer [24].

While trastuzumab deruxtecan is the only antibody-drug conjugate (ADC) approved currently for HER-2 positive gastroesophageal cancer, there are several agents in development with different cytotoxic payloads and linker molecules to optimize the efficacy and minimize adverse events [25]. PF-05804103 is an ADC that includes a trastuzumab-derived antibody and a derivative of auristatin, which has demonstrated promising efficacy in preclinical studies [26]. Preliminary study of a phase I study (NCT03284723) of HER2-positive solid tumors including breast and gastroesophageal cancers demonstrate an ORR of 52.4% (11/21 patients) [27]. Another promising ADC is XMT-1522, which contains a human IgG1 anti-HER2 monoclonal antibody linked with the auristatin derivative (AF-HPA). Preliminary results from a phase I clinical trial which enrolled 19 patients with HER2-expressing breast, lung, and gastric tumors demonstrated a disease-control rate of 83% (5/6 patients) with 1 PR and 4 SD [28]. These innovative agents combine the specificity of an antibody along with a potent cytotoxic payload, with the promise of offering a favorable therapeutic index that may not only benefit patients with chemorefractory disease but may also one day be used in earlier lines of therapy.

Enrichment and homogeneity of HER2 amplification is critical to conventional HER2-based therapies; however, intratumoral heterogeneity of HER2 expression within gastroesophageal cancer may be a mechanism of resistance to anti-HER2 therapy and limit clinical efficacy to these agents [28,29]. Furthermore, recent molecular-sequencing efforts by The Cancer Genome Atlas (TCGA) have highlighted the molecular heterogeneity between patients by identifying four molecular subtypes of gastric cancer: microsatellite instable (MSI), Epstein–Barr virus (EBV)-associated, chromosomally instable (CIN), and genomically stable (GS) tumors [30]. The molecular diversity between patients, along with intratumoral heterogeneity, highlight the complexities of managing this disease entity and reinforce the importance of harnessing genomic sequencing to understanding disease biology and guiding clinical decision making.

Next-generation sequencing (NGS) may play a complementary role to immunohistochemistry/FISH to identify patients that may benefit from anti-HER therapy. A recent study of the MSK-IMPACT NGS assay demonstrated an overall 98.4% concordance with traditional IHC/FISH HER2 testing [31]. Discrepancies were largely attributed to intratumoral heterogeneity and low tumor sample. Furthermore, studies have demonstrated that high levels of HER2 identified by NGS are associated with response to anti-HER2 therapy, which supports its use as a tool in clinical practice [32]. Additionally, NGS may also identify genetic coalterations that impact the efficacy of anti-HER2 agents. A recent study of 44 patients with post-trastuzumab tissue samples analyzed by NGS identified several genes that predicted resistance to anti-HER therapies, including mutations in *KRAS*, *PI3K* signaling pathways, and exon 16 deletion of the HER2 gene [33]. Tumors harboring coalterations within *RTK*-*RAS*-*PI3K*-*AKT* signaling pathways demonstrated a shorter median PFS of 8.4 months, while tumors highly enriched for HER2 were found to have significantly prolonged PFS of 24.3 months. These findings suggest that NGS can not only identify tumors that are enriched with HER2, but also identify genetic coalterations that further refine our understanding of the disease biology and enhance our ability to identify patients who would derive the greatest benefit from anti-HER2 therapy.

In addition to genomic sequencing of tumor tissue, circulating tumor DNA (ctDNA) is another promising assay that may complement NGS and IHC/FISH to better understand the molecular and genomic diversity of gastroesophageal cancer. Several large retrospective studies of solid tumors, including gastric cancer, have found that ctDNA was able to detect clinically relevant genetic alterations in the Guardant360 and FoundationACT platforms at a rate of 85% and 82%, respectively from corresponding matched tissue samples [34,35]. While ctDNA was able to identify genetic aberrations found in matched tissue NGS, ctDNA was able to identify additional alterations not seen in the corresponding tissue sample, which likely reflects the broad tumoral molecular heterogeneity. In a companion biomarker study to the prospective T-ACT clinical trial, ctDNA was able to identify the HER2 gene amplification in 41/68 patients (60.3%) [36]. Additionally, in another parallel biomarker ctDNA study of a prospective phase II study of lapatinib with capecitabine and oxaliplatin in advanced HER2-positive gastric adenocarcinoma, HER2 copy-number amplification identified by liquid biopsies correlated strongly with response to anti-HER therapy [32]. While these studies are promising, large-scale prospective studies will be needed to validate the composite use of NGS and ctDNA in conjunction with IHC/FISH as a guide to clinical decision making. Furthermore, future studies should also focus on exploring the role of using these assays to understand how the cancer’s genomic profile evolves over time in response to therapy and genetic instability to better understand the mechanisms of resistance to therapy. Nonetheless, ctDNA is a useful adjunct, particularly in scenarios when a repeat biopsy is not feasible, or there is not enough tissue sample to perform next-generation sequencing.

Several promising clinical trials under study are likely to reshape our understanding and approach to HER2-positive gastroesophageal cancer in the coming years. A phase II/III clinical study of evorpacept (ALX148), a CD47 inhibitor, (ASPEN-06; NCT NCT05002127) is currently actively recruiting patients after promising results from its phase I counterpart. ASPEN-01 was a phase I study that evaluated the role of evorpacept in combination with trastuzumab, ramucirumab, and paclitaxel as second-line therapy for patients with advanced HER2-positive gastroesophageal cancer. In this study of 18 patients, evorpacept in combination with trastuzumab and chemotherapy demonstrated a robust objective response rate of 72% and estimated overall survival of 76% at 12 months. These results compare favorably to randomized historical control studies, such as RAINBOW with ORR of 28% and OS at 12 months of 40%, and DESTINY-gastric 01 with ORR of 41% and OS at 12 months of 52% [37,38]. Preliminary data suggest that this combination is well-tolerated, with an acceptable safety profile. Given these findings, the U.S. Food and Drug Administration (FDA) recently granted orphan drug designation to evorpacept on 27 January 2022 for the treatment of patients with gastroesophageal cancer. Another promising therapeutic currently under the phase 2/3 clinical trial (MAHOGANY, NCT NCT04082364) is the anti-HER2 monoclonal antibody margetuximab in combination with retifanlimab, a PD-1 inhibitor, or tebotelimab, which binds PD-1 and lymphocyte activation gene 3 concomitantly, with or without chemotherapy as first-line therapy for HER2-positive, PD-L1-positive gastroesophageal adenocarcinoma [39]. Interim data from cohort A (margetuximab with retifanlimab) of the MAHOGANY study presented at ESMO 2021 found that 21 of 40 patients (53%) achieved an objective response, with four patients achieving a complete response. The median duration of response was 10.3 months. Overall, the treatment regimen was well-tolerated, with grade ≥ 3 adverse events in 19% of patients. These promising findings, if validated, will potentially offer these patients a chemotherapy-free combination for first-line therapy in HER2-positive GEA. Another regimen of interest is tucatinib, trastuzumab, ramucirumab, and paclitaxel currently under active investigation in the phase II/III MOUNTAINEER-02 study (NCT04499924) after promising results from the phase I study for metastatic colorectal cancer patients [40,41]. In the initial phase I study of 22 evaluable patients, the authors reported an objective response rate of 55% with a median OS was 17.3 months while PFS was 6.2 months. Given these promising results, the highly selective nature of tucatinib for HER2-positive disease, and favorable side-effect profile, this new study seeks to evaluate this regimen for patients with advanced HER2-positive gastroesophageal cancer who previously received HER2-directed antibody and received one prior line of therapy. The phase II study will find the optimal dosage of paclitaxel to be used given potential impact of tucatinib on paclitaxel metabolism while the primary outcomes in the phase II study will evaluate median OS and PFS. Other active phase III clinical studies under study include the DESTINY-gastric04 (NCT04704934), which seeks to validate the role of trastuzumab deruxtecan in comparison to ramucirumab and paclitaxel for patients with advanced HER2-positive gastroesophageal cancer who progressed with a trastuzumab-based regimen. The primary outcomes in this study will be median OS and PFS. This study is currently actively recruiting patients.

## 5. Conclusions

In conclusion, while significant challenges remain to the management of HER2-positive gastroesophageal cancer, there are also promising diagnostic and therapeutic advances on the horizon. Many innovative agents, including antibody-drug conjugates, are currently in development for a heavily pretreated population; however, future studies will help assess whether these therapies may be offered earlier to patients and even potentially in early-stage disease. Future studies will also help clarify the role of next-generation sequencing and circulating-tumor DNA to better understand the disease biology and molecular diversity and heterogeneity both within the tumor and between patients. Furthermore, these tools may not only help guide clinical decision making, but also identify new gaps in our understanding to develop the next generation of therapeutic agents for this disease.

## Figures and Tables

**Table 1 diseases-10-00023-t001:** First-line seminal clinical trials of HER2 targeted therapies in HER2-positive advanced gastroesophageal cancer.

*n* (Phase)	Treatment Setting	Experimental Arm	Control Arm	Primary Endpoint	Results	Ref.
594 (phase III)	First-line	Chemotherapy (capecitabine/cisplatin or 5-fluorouracil/cisplatin) + trastuzumab	Chemotherapy alone	OS	13.8 vs. 11.1 months (HR 0.74; 95% CI 0.60–0.91; *p* = 0.0046)	[6]
248 (phase III)	First-line	Chemotherapy (cisplatin/capecitabine) + trastuzumab (8 mg/kg loading dose, followed by 6 mg/kg vs. 10 mg/kg every 3 weeks)	Chemotherapy + trastuzumab (10 mg/kg every 3 weeks)	OS	12.5 vs. 10.6 months (HR 1.24; 95% CI, 0.86–1.78; *p* = 0.2401)	[7]
545 (phase III)	First-line	Chemotherapy (capecitabine/oxaliplatin) + lapatinib	Chemotherapy alone	OS	12.2 vs. 10.5 months (HR 0.91; 95% CI, 0.73–1.12)	[8]
780 (phase III)	First-line	Chemotherapy (cisplatin, capecitabine, or 5-fluorouracil) + trastuzumab/pertuzumab	Chemotherapy + trastuzumab	OS	17.5 vs. 14.2 months (HR = 0.84, *p* = 0.057)	[10]
43 (phase IB/II)	First-line	Chemotherapy (capecitabine + cisplatin) + trastuzumab + pembrolizumab	None	ORR	76.7% (CR 16.3%, PR 60.5%, conversion surgery 4.6%)	[11]
264 (phase III)	First-line	Chemotherapy (5-flurouracil/cisplatin or capecitabine/oxaliplatin) + trastuzumab/pembrolizumab	Chemotherapy + trastuzumab	Interim analysis: ORR	74% vs. 52% (one-sided *p*-value < 0.0001, statistically significant)	[12]

PD, progressive disease; ORR, overall response rate; CI, confidence interval; OS, overall survival; PFS, progression-free survival; HR, hazard ratio.

**Table 2 diseases-10-00023-t002:** Second-line and beyond seminal clinical trials of HER2-targeted therapies in HER2-positive advanced gastroesophageal cancer.

*n* (Phase)	Treatment Setting	Experimental Arm	Control Arm	Primary Endpoint	Results	Ref.
415 (phase II/III)	Second-line	Trastuzumab emtansine (T-DM1)	Intravenous docetaxel 75 mg/m^2^ every 3 weeks or intravenous paclitaxel 80 mg/m^2^ weekly	OS	7.9 vs. 8.6 months (HR 1.15; 95% CI 0.87–1.51, one-sided *p* = 0.86)	[15]
273 (phase III)	Second-line	Lapatinib 1500 mg daily plus weekly paclitaxel 80 mg/m^2^ or paclitaxel alone	Paclitaxel	OS	11.0 vs. 8.9 months (HR 0.84; 95% CI 0.64–1.11; *p* = 0.1044)	[17]
91 (phase II)	Second-line	Paclitaxel + trastuzumab	Paclitaxel	PFS	3.7 vs. 3.2 months (HR 0.91; 80% CI, 0.67–1.22; *p* = 0.33)	[18]
187 (phase II)	Third-line	Trastuzumab deruxtecan	Physician’s choice of chemotherapy	ORR	51% vs. 14% (*p* < 0.001)	[19]
79 (phase II)	Second-line	Trastuzumab deruxtecan	None	ORR	38% (95% CI 27.3–49.6)	[20]

PD, progressive disease; ORR, overall response rate; CI, confidence interval; OS, overall survival; PFS, progression-free survival; HR, hazard ratio.

## Data Availability

Not applicable.

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
