# Peer review of "Therapeutic Advances and Challenges in the Management of HER2-Positive Gastroesophageal Cancers"

_diseases, 2022, doi:10.3390/diseases10020023_

Round 1
Reviewer 1 Report
Review: Therapeutic advances and challenges in the management of HER2-positive gastroesophageal cancers
Authors affiliations: footnote 10 does not appear in authors information
Spelling:
Page 2 line 98: double past tense (“did demonstrated”)
Table 2: headline: different formats
Supplemental material, funding, authors conflict of interest: MISSING!
Contents: The authors provide a substantial overview on HER2 targeting in gastroesophageal cancers, with major focus on oncological studies and treatment protocols. I appreciate that methods of HER2 analysis are mentioned and even scores are given as important therapy guiding stratifiers and most essentially, predictive marker.
Here, I was hoping (from mentioning in the abstract) that analysis methods like NGS, cut-offs, correlation of FISH and immunohistochemistry, type of immunohistochemistry clones and most importantly scoring methods were given more attention. NGS and ct DNA is only mentioned in the discussion, not in the results section. Do recent studies exist ? Whole genome sequencing studies?
As was shown for HER2 in colon cancer, different HER2 clones and scoring systems vary in the correlation with (gold standard) fluorescence in Situ hybridization (FISH). Does this apply to gastroesophageal cancers?
Page 3 line 104 abbreviations like q3w should be explained to non-oncologists
Page 6, line 262: Reference missing for statement of heterogeneity
General consideration: The discussion seems more like a further results section. Here, a study for NGS approaches is even mentioned! The discussion should be much more concise, pinpointing controversies and for instance pros and cons of HER2 status guided therapies (attempt given) or clinical outcome of heterogeneous cases.
Author Response
Thank you for taking the time to thoroughly review this article and providing thoughtful comments.
- Author attributions reviewed and corrections made.
- Spelling and grammatical mistakes fixed.
- Supplemental material, funding, authors conflict of interest added.
- Q3W expanded for clarity.
- Appropriate reference for tumoral heterogeneity included.
- Due to the emerging picture of tumoral heterogeneity of HER2 positive in particular and gastroesophageal cancers in general, larger datasets will be needed to ascertain if composite testing with traditional IHC/FISH, tumor NGS, and ctDNA, as well as other biomarkers, can enrich for patients that derive the most clinical benefit in HER2-directed strategies. The data for NGS/ctDNA and other biomarkers remains limited and our decision to include this in the discussion was to highlight the urgent need for additional research in this space. We agree that these ancillary diagnostic studies are highly important for clinical decision-making however we decided to emphasize the review on the significant changes in therapeutic landscape for HER2 GEA.
Reviewer 2 Report
Manuscript entitled "Therapeutic advances and challenges in the management of HER2-positive gastroesophageal cancers". This work is well-organized and sound. Some issues as follows should be addressed before final acceptance:
- The predictive marker for the clinical efficacy of Her-2 targeting should be listed.
- The positive rate defined by IHC, FISH, and NGS for Her-2 positive in various gastroesophageal cancers should be listed.
Author Response
Thank you for taking the time to thoroughly review this article.
- You raise a really great point. Some examples of emerging tumor markers we highlight in the discussion section include how NGS may predict co-alterations that may be useful biomarkers that predict resistance to anti-HER2 therapy. Due to the emerging picture of tumoral heterogeneity of HER2 positive in particular and gastroesophageal cancers in general, larger datasets will be needed to ascertain if composite testing with traditional IHC/FISH, tumor NGS, and ctDNA, as well as other biomarkers, can enrich for patients that derive the most clinical benefit in HER2-directed strategy.
- That is a great thought. I have included a statement on how HER2-positivity can vary depending on anatomic location of the disease and the histologic subtype of the tumor.
Reviewer 3 Report
This is a very short and "regular" review on therapeutic advances in HER-2 positive GE cancer. The paper has many irregularities in terms of character style, increased blank spaces, change of character into the same paragraph.
I think the review lacks of originality, a figure is missing. I would have described in a single paragraph the ongoing studies with an additional table also. The impression is that this is a fast written paper that could improve with longer and more precise work.
Author Response
Thank you for taking the time to thoroughly review this paper.
Our hope was that by incorporating the expertise from surgical oncology, medical oncology, radiation oncology, and pathology we could bring the unique perspectives of the different specialties and present a cohesive and concise evidence-based review on the diagnostic and therapeutic landscape for HER2-gastroesophageal cancers. This was a significant undertaking that took several months to complete in a thoughtful fashion. We decided to discuss in significant detail ongoing phase 3 clinical trials recognizing that many of these studies may lead to changes in the treatment paradigm in the near future and yield other novel research questions to address.
Round 2
Reviewer 2 Report
It appears that the authors made some modification and this work is acceptable now.
Reviewer 3 Report
Thanks for your response. My evaluation remains the same. The review has no point of novelty.